# The Furin Protease Dependence and Antiviral GBP2 Sensitivity of Murine Leukemia Virus Infection Are Determined by the Amino Acid Sequence at the Envelope Glycoprotein Cleavage Site

**DOI:** 10.3390/ijms25189987

**Published:** 2024-09-16

**Authors:** Yoshinao Kubo, Manya Bakatumana Hans, Taisuke Nakamura, Hideki Hayashi

**Affiliations:** 1Department of Clinical Medicine, Institute of Tropical Medicine, Nagasaki University, Nagasaki 852-8523, Japan; manyabakat@gmail.com (M.B.H.); nakamurat@nms.ac.jp (T.N.); 2Program for Nurturing Global Leaders in Tropical Medicine and Emerging Communicable Diseases, Graduate School of Biomedical Sciences, Nagasaki University, Nagasaki 852-8523, Japan; 3Medical University Research Administration, Nagasaki University School of Medicine, Nagasaki 852-8523, Japan; hhayashi@nagasaki-u.ac.jp

**Keywords:** murine leukemia virus, envelope glycoprotein, GBP2, furin

## Abstract

Host restriction factor GBP2 suppresses the replication of the ecotropic Moloney murine leukemia virus (E-MLV) by inhibiting furin protease, which cleaves the viral envelope glycoprotein (Env) into surface (SU) and transmembrane (TM) subunits. We analyzed the impacts of GBP2 on the infection efficiency mediated by MLV Envs of different strains of ecotropic Moloney, polytropic Friend, amphotropic, and xenotropic MLV-related (XMRV) viruses. Interestingly, the Envs of ecotropic Moloney and polytropic Friend MLV were sensitive to the antiviral activity of GBP2, while XMRV and amphotropic Envs showed resistance. Consistent with the sensitivity to GBP2, the amino acid sequences of the sensitive Envs at the SU-TM cleavage site were similar, as were the sequences of the resistant Envs. SU-TM cleavage of the GBP2-sensitive Env protein was inhibited by furin silencing, whereas that of GBP2-resistant Env was not. The substitution of the ecotropic Moloney cleavage site sequence with that of XMRV conferred resistance to both GBP2 and furin silencing. Reciprocally, the substitution of the XMRV cleavage site sequence with that of the ecotropic sequence conferred sensitivity to GBP2 and furin silencing. According to the SU-TM cleavage site sequence, there were sensitive and resistant variants among ecotropic, polytropic, and xenotropic MLVs. This study found that the dependence of MLV Env proteins on furin cleavage and GBP2-mediated restriction is determined by the amino acid sequences at the SU-TM cleavage site.

## 1. Introduction

Murine leukemia virus (MLV) is a prototype belonging to the gammaretrovirus family and is utilized as a mouse model of various human disorders, including leukemia [1], immunodeficiency [2,3,4], and neurological diseases [5]. Additionally, an MLV-derived vector is widely used as a tool to transfer a gene of interest into target cells in many fields of basic and medical biology [6]. Because an MLV vector is typically constructed through the transfection of human 293T cells, understanding the molecular mechanisms of MLV replication in these cells is crucial for developing an improved MLV vector with enhanced efficiency in gene transfer.

The MLV encodes only three structural proteins, named Gag, Pol, and Env. The Gag protein is initially synthesized as a precursor and subsequently cleaved into the matrix, p12, capsid, and nucleocapsid components by the viral protease. The Pol protein is translated as a fusion precursor with the Gag protein and is then enzymatically processed into protease, reverse transcriptase, and integrase by its protease activity. The Env protein is also produced as a precursor glycoprotein, which is cleaved by a cellular protease into surface (SU) and transmembrane (TM) subunits [7].

The retrovirus replication cycle can be succinctly divided into two main phases: the early and late phases. The early phase encompasses several key steps, including the recognition of a cell surface receptor, entry into the host cell cytoplasm through fusion between the viral and host cell membranes, reverse transcription of the viral RNA genome into DNA, and integration of the viral DNA into the host cell chromosomal DNA. Meanwhile, the late phase involves the expression of viral genes from the integrated genome, assembly of viral RNA and proteins, budding of immature viral particles from the infected host cells, and maturation of viral particles through processing of the viral proteins.

It is thought that the host’s furin protease mediates the SU-TM cleavage of the MLV envelope glycoprotein (Env) [8]. The Env precursor’s cleavage of MLV is essential for viral replication. The trafficking of Env mutants that fail to be cleaved into SU and TM is impaired, and the precursor proteins are not incorporated into virions [8,9,10,11]. After cleavage, trimers of SU-TM heterodimers are formed on the cell surface and incorporated into viral particles. The SU subunit interacts with a cell surface receptor, triggering a conformational change in the TM subunit, thereby activating its membrane fusion activity.

The SU subunit determines species specificity, delineated into ecotropic, amphotropic, polytropic, and xenotropic strains [12]. Ecotropic MLV (E-MLV) can infect mouse cells, with its SU recognizing the cationic amino acid transporter 1 (CAT1) as the cell surface receptor [13,14,15]. Amphotropic MLV (A-MLV) exhibits a broad host range, infecting various mammals, including humans, minks, rats, and mice. Its SU subunit binds to phosphate transporter 2 (Pit2) [16,17,18]. Polytropic and xenotropic MLVs (P-MLV and X-MLV, respectively) both engage with the XPR1 cell surface protein via the SU protein [19,20,21,22], although its physiological function remains unresolved [23,24]. P-MLV shares a wide range of hosts with A-MLV. Although X-MLVs are isolated from mice, it fails to infect laboratory mouse cells but exhibits infectivity in other mammalian cells.

Host cells possess numerous restriction factors to impede retroviral replication, thus safeguarding against retroviral infections [25,26,27]. Upon the detection of exogenous viruses, host cells initiate the expression of the type I interferon (IFN) [28]. Moreover, certain cells such as T lymphocytes and macrophages secrete type II IFN upon stimulation by pathogens. These IFNs play pivotal roles in activating the expression of various host antiviral restriction factors. These host antiviral factors exhibit specificity in targeting different steps of the retroviral replication cycle. For example, members of the guanylate binding protein (GBP) family, GBP2 and GBP5, function by impeding retroviral replication through the downregulation of furin, a cellular protease responsible for cleaving the viral Env glycoprotein into SU and TM subunits [29,30,31]. There are many lines of evidence showing the inhibitory effects of these GBP family members against other viruses [32,33,34,35,36,37,38,39,40,41,42,43].

Previously, we identified host factors, including GBP2, whose expressions are upregulated by IFN-γ, a type II IFN, using microarray analysis [44]. We then conducted a comprehensive analysis to assess the impact of these host factors on transduction by the A-MLV-pseudotyped human immunodeficiency virus type 1 (HIV-1)-based vector. Finally, we discovered that the expression of FAT10, IDO1, IFI6, and GILT in target cells restricted transduction, showing the inhibition of the early phase in the retroviral life cycle by these host factors [44]. It was speculated that a few factors might also inhibit the late phase, and further investigation was necessary to confirm this speculation. Therefore, we evaluated the effects of these factors on the late phase of the replication cycle. However, the results revealed that none of these factors inhibited the late phase.

It has been previously documented that GBP2 hinders E-MLV replication by inducing furin degradation [31]. However, our study showed that GBP2 did not influence transduction by an A-MLV-pseudotyped HIV-1 vector. Consequently, we investigated the impacts of GBP2 on the infection by MLVs exhibiting different strains in this study.

## 2. Results

### 2.1. GBP2 Does Not Inhibit Transduction by Amphotropic MLV-Pseudotyped HIV-1 Vector

We previously identified and cloned cDNAs of 11 human factors, including GBP2, whose expressions are upregulated by IFN-γ using microarray analysis [44]. To investigate the impact of these factors on the late phase of the retrovirus replication cycle, HeLa cells stably expressing these host factors were generated via VSV-pseudotyped MLV vectors encoding the factors, as previously described [44]. The transduced HeLa cells were then transfected with three plasmids to construct a luciferase-expressing A-MLV-pseudotyped HIV-1 vector. Subsequently, culture supernatants of the transfected cells were inoculated into human TE671 cells to measure transduction titers. None of the factors showed any significant effect on titers of the pseudotyped vector, when compared to those from cells transduced with an empty vector (Figure 1). These results indicate that none of the factors tested has an impact on the late phase of the retroviral replication cycle.

### 2.2. X-MLV and A-MLV Are Resistant to GBP2, but E-MLV and P-MLV Are Susceptible

Although in our study the titers of A-MLV-pseudotyped HIV-1 vector produced from GBP2-expressing cells were comparable to those from control cells, previous reports have demonstrated that GBP2 expression in Moloney E-MLV-producing cells inhibits its replication by downregulating furin [31]. Hence, the distinct strains of MLVs may result in differing susceptibility to the antiviral activity of GBP2. To assess this hypothesis, we examined the effects of GBP2 expression in cells producing replication-defective ecotropic (E-MLV), polytropic (P-MLV), amphotropic (A-MLV), or xenotropic (X-MLV) vectors on their transduction titers. For this analysis, we utilized Env expression plasmids derived from Moloney ecotropic MLV [45], Friend mink cell focus forming MLV [46], and xenotropic murine leukemia virus-related virus [47] to construct replication-defective E-MLV, P-MLV, and X-MLV vectors, respectively. Only a single clone of amphotropic MLV (A-MLV) is available [48], and an expression plasmid of A-MLV Env protein was previously generated in our research [49].

Human 293T cells were transfected with expression plasmids to generate luciferase-expressing E-MLV, P-MLV, A-MLV, or X-MLV vectors, along with either pcDNA3.1 or an N-terminally GFP-tagged GBP2 (GFP-GBP2) expression plasmid. The culture supernatants of the transfected cells were then inoculated into human TE671 cells expressing the E-MLV receptor CAT1 (TE671/CAT1) to assess transduction titers. Interestingly, the transduction titer from cells transfected with GFP-GBP2 was lower than in those transfected with the pcDNA3.1 for both E-MLV and P-MLV vectors (Figure 2A). However, transduction titers from cells transfected with either pcDNA3.1- or GFP-GBP2 were similar for A-MLV and X-MLV vectors. This result suggests that E-MLV and P-MLV are susceptible to GBP2-mediated antiviral activity, while A-MLV and X-MLV demonstrate resistance.

The amino acid sequences at the SU-TM cleavage site of E-MLV, P-MLV, A-MLV, and X-MLV Env proteins were compared (Figure 2B), because GBP2 inhibits E-MLV replication by downregulation of furin expression. The amino acid sequences of GBP2-susceptible E-MLV and P-MLV Env share the sequence SXRHKR, where X represents an uncharged hydrophilic residue N or Y, while the sequences of GBP2-resistant A-MLV and X-MLV Env are characterized by BTKYKR with B indicating a positively charged basic residue K or R. Notably, X-MLV Env exhibits only a single distinct amino acid residue compared to E-MLV within the region excluding the BTKYKR sequence. Consequently, in subsequent experiments, E-MLV and X-MLV were employed.

To investigate the impact of GFP-GBP2 on the digestion of MLV Env glycoprotein to SU and TM subunits, 293T cells were transfected with either the E-MLV or X-MLV Env expression plasmid along with a pcDNA3.1 or GFP-GBP2 expression plasmid. Cell lysates from the transfected cells were then subjected to Western blot analysis. Interestingly, while the levels of Env precursor (SU-TM) remained similar regardless of GFP-GBP2 presence, cells transfected with E-MLV Env and GFP-GBP2 exhibited lower levels of mature TM than those transfected with E-MLV Env and pcDNA3.1 (Figure 2C). Conversely, the presence of GFP-GBP2 did not alter the level of cleaved TM of X-MLV Env. These results indicate that GBP2 inhibits the SU-TM digestion of E-MLV Env but not that of X-MLV Env. Collectively, it was found that the expression of GBP2 in vector-producing cells inhibits the E-MLV Env-mediated transduction by suppressing SU-TM digestion, while X-MLV Env remains resistant to GBP2.

### 2.3. The Susceptibility of GBP2 Is Determined by the Amino Acid Sequence at the SU-TM Cleavage Site

To investigate whether the amino acid sequence near the SU-TM cleavage site determines the susceptibility to GBP2, a mutant of E-MLV Env substituting the sequence with X-MLV Env was constructed by site-directed mutagenesis (Table 1), and named E-MLV FM (Figure 3A). To investigate the effect of GBP2 on the transduction by the E-MLV FM mutant, 293T cells were transfected with expression plasmids to construct a luciferase-encoding MLV vector containing E-MLV Env wild type (Wt) or FM mutant along with a pcDNA3.1 or GFP-GBP2 expression plasmid. The culture supernatants of the transfected cells were inoculated into TE671/CAT1 cells, and the luciferase activity of the inoculated cells was measured. Consistent with the above result, transduction mediated by E-MLV Env Wt was reduced by GFP-GBP2 but not that by E-MLV FM mutant Env like X-MLV Env Wt (Figure 3B). This result shows that E-MLV Wt is susceptible to GBP2 but E-MLV FM mutant is resistant.

To investigate the impact of GBP2 on SU-TM cleavage of the E-MLV FM mutant Env, cell lysates from the transfected 293T cells were subjected to Western blot analysis. The level of mature TM in cells transfected with the E-MLV Env Wt was decreased by GFP-GBP2 but not those with the E-MLV FM mutant (Figure 3C). To confirm the result, ratios of mature TM to precursor protein were calculated. As expected, the ratios of E-MLV Wt were reduced by GFP-GBP2, but those of E-MLV FM mutant were not (Figure 3D). These results indicate that SU-TM cleavage of the E-MLV Env Wt is inhibited by GBP2, but that of the E-MLV Env FM mutant is resistant like the X-MLV Env Wt.

Additionally, we generated another mutant of X-MLV that has substitution by the region at the SU-TM cleavage site of E-MLV and designated it as X-MLV FM mutant (Figure 4A). To assess the impact of GBP2 on transduction mediated by the X-MLV Env Wt or FM mutant, 293T cells were transfected with the expression plasmids to construct a luciferase-encoding MLV vector that contains X-MLV Env Wt or FM mutant, along with a pcDNA3.1 or GFP-GBP2 expression plasmid. The culture supernatants of the transfected cells were inoculated into TE671 cells, and luciferase activity of the inoculated cells was measured. Luciferase activity by the X-MLV Env Wt was not affected by GFP-GBP2, but that by the X-MLV FM mutant was reduced by GFP-GBP2 (Figure 4B). This result demonstrates that the X-MLV Env Wt is resistant to GBP2, but the X-MLV FM mutant is susceptible like the E-MLV Env Wt.

To know the impact of GBP2 on the SU-TM cleavage of X-MLV Env Wt and FM mutant, cell lysates from the transfected 293T cells were subjected to Western blot analysis. Ratios of mature TM to precursor in the X-MLV Env Wt were not changed by GFP-GBP2 but were decreased in the FM mutant by its expression (Figure 4C,D). This result shows that the SU-TM cleavage of the X-MLV Env FM mutant is inhibited by GBP2 like the E-MLV Env Wt, but that of the X-MLV Env Wt is not.

Mouse GBP2 did not have this function.

Taken together, as expected, it was found that transduction and SU-TM cleavage of the E-MLV Env protein containing the SXRHKR sequence at the SU-TM cleavage site are susceptible to GBP2, and those of the X-MLV Env containing the BTKYKR sequence are resistant to GBP2.

### 2.4. The Susceptibility to Furin Silencing Is Determined by the Amino Acid Sequence at the SU-TM Cleavage Site

The subsequent investigation focused on evaluating the impact of furin silencing on transduction by the E-MLV vector, as GBP2 inhibits E-MLV replication by downregulating furin protease. Human 293T cells were transduced with a VSV-pseudotyped HIV vector expressing an shRNA targeting furin mRNA. Following transduction, the 293T cells were subjected to puromycin treatment, because the HIV vector also carries the puromycin-resistant gene. The resulting puromycin-resistant cell pool (293T/shFurin) was utilized in subsequent experiments. In parallel, 293T cells were transduced by a VSV-pseudotyped HIV vector containing a scrambled sequence, and the resulting puromycin-resistant cell pool (293T/shCont) was used in subsequent experiments as a control. The levels of furin protein in 293T/shFurin cells were indeed lower than those in 293T/shCont cells (Figure 5A).

The expression plasmids to construct luciferase-encoding MLV vector containing either E-MLV Env Wt or FM mutant were transfected into 293T/shCont and 293T/shFurin. The culture supernatants from the transfected cells were inoculated into TE671/CAT1 cells. Notably, the transduction titers of the E-MLV Wt vector produced from 293T/shFurin cells were found to be lower than those from 293T/shCont cells (Figure 5B). Conversely, the transduction titers of the E-MLV FM mutant vector showed no significant difference between 293T/shCont and 293T/shFurin cells.

The cell lysates prepared from the transfected cells were subjected to Western blot analysis. The mature TM levels of E-MLV Env Wt in 293T/shFurin cells were lower than those in 293T/shCont cells (Figure 5C,D). However, there was no significant change in the mature TM levels of the FM mutant between the two cell types. The ratios of mature TM to precursor in E-MLV Env Wt were reduced upon furin silencing, whereas this effect was not observed in the E-MLV Env FM mutant. These findings suggest that furin silencing impedes transduction mediated by the E-MLV Env Wt by suppressing SU-TM cleavage. In contrast, furin silencing did not affect transduction mediated by the E-MLV Env FM mutant. Thus, the results indicate that furin silencing selectively inhibits E-MLV Env Wt-mediated transduction by disrupting SU-TM cleavage, while leaving E-MLV FM mutant-mediated transduction unaffected.

To explore the influence of furin silencing on transduction and SU-TM cleavage of X-MLV Env Wt and FM, 293T/shCont and 293T/shFurin cells were transfected with expression plasmids to construct a luciferase-encoding MLV vector containing X-MLV Env Wt or FM mutant. Subsequently, the culture supernatants from the transfected cells were used to inoculated TE671 cells, and the luciferase activities of the inoculated cells were measured. The luciferase activities of the X-MLV Wt vector produced from transfected 293T/shCont cells were comparable to those from 293T/shFurin cells (Figure 6A). However, transduction by the X-MLV Env FM mutant was reduced by furin silencing. Moreover, furin silencing inhibited the SU-TM cleavage of X-MLV Env FM mutant, while having no significant effect on that of X-MLV Env Wt (Figure 6B,C).

Taken together, these findings indicate that both transduction and SU-TM cleavage of the X-MLV Env protein containing the SXRHKR sequence at the SU-TM cleavage site are diminished by furin silencing, whereas those of the X-MLV Env containing the BTKYKR sequence remain unaffected.

### 2.5. Amino Acid Sequences at the SU-TM Cleavage Sites of Different MLVs

The above results reveal that E-MLV and P-MLV are susceptible to GBP2-mediated antiviral activity, whereas A-MLV and X-MLV display resistance. Specifically, the amino acid sequences SXRHKR and BTKYKR adjacent to the SU-TM cleavage site confer susceptibility and resistance to GBP2, respectively. To further explore whether MLV tropism consistently correlates with GBP2 susceptibility, the amino acid sequences near the cleavage sites across various MLV strains were compared.

As shown in Figure 7, the Friend E-MLV Env protein [50] features the SNRHKR sequence, while the BM5eco [51] and Akv [52] E-MLV Env proteins exhibit RAKYKR. This implies that Friend E-MLV, akin to Moloney E-MLV, is susceptible to GBP2, contrasting with BM5eco and Akv E-MLV. Similarly, Moloney P-MLV [53] contains the SNRHKR sequence, whereas PTV1 and CasE#1 P-MLV [54] possess RAKYKR and KTKYKR, respectively. This suggests susceptibility of Moloney P-MLV, akin to Friend P-MLV, in contrast to PTV1 and CasE#1 P-MLVs. Furthermore, EKVX [55] and Bxv-1 [56] X-MLV Env proteins harbor the KTKYKR and SYRHKR sequences, respectively, indicating distinct susceptibility to GBP2. These findings underscore that GBP2 susceptibility is dictated by the amino acid sequence at the SU-TM cleavage site rather than tropism. Other sequence groups were not detected.

## 3. Discussion

It was discovered that MLV Env proteins harboring the SXRHKR amino acid sequence at the SU-TM cleavage site undergo digestion by furin, and their entry is hindered by GBP2. Conversely, those containing BTKYKR are cleaved by cellular protease(s) distinct from furin, rendering their entry impervious to GBP2-mediated inhibition. The dependence on furin and susceptibility to GBP2 are determined by the amino acid motif at the SU-TM cleavage site rather than MLV tropism.

MLV Env proteins containing BTKYKR, where B denotes R or K basic residue, are likely to be targeted for digestion by proprotein convertases including furin. Both sequences feature three to four basic amino acids within a span of six residues, a characteristic recognized by furin and other proprotein convertases for substrate protein digestion [57,58]. Hence, the BTKYKR Env is anticipated to be cleaved by other proprotein convertases. However, it should be noted that this observation does not entirely rule out the possibility of the BTKYKR Env cleavage by furin; it remains plausible that it may undergo digestion by cellular proteases including furin and other proprotein convertases. The GBP2-resistant Env, while typically cleaved by furin in wild-type cells, exhibits an altered fate when furin is suppressed by shRNA or GBP2. In such cases, the Env becomes susceptible to digestion by other proteases. To identify the protease that digests the BTKYKR Env protein, further study is required. In contrast, the SXRHKR-containing Env protein is exclusively cleaved by furin. Consequently, when furin is knocked down, the entry mediated by the SXRHKR Env is impeded.

MLV Env proteins have acquired sequence heterogeneity at the SU-TM cleavage sites as a strategy to confer resistance against host defense factors. These proteins can be categorized into two groups based on the amino acid sequences at the SU-TM cleavage site. The emergence of this heterogeneity can be attributed to the evolutionary arms race between hosts and viruses. Hosts have evolved numerous antiviral factors to defend against viral infections. In response, viruses have developed mechanisms to evade or counteract these host defense factors, often manifesting as sequence heterogeneity within viral genomes. For example, HIV-1 has been observed to exhibit sequence diversity as means to circumvent host APOBEC3-mediated antiviral activity [59]. Similarly, certain variants of SARS-CoV-2 have been reported to resist GBP2 through substitutions at the cleavage site of its spike protein [33]. Thus, it is plausible that the heterogeneity observed in MLV at the SU-TM cleavage site has arisen to confer resistance against GBP2. In response to the viral strategies aimed at overcoming GBP2-mediated antiviral activity, hosts may have diversified their repertoire of proprotein convertases, potentially representing a countermeasure against viral evasion tactics.

Mouse GBP2 did not demonstrate inhibition of entry mediated by MLV Env protein in mouse NIH3T3 cells, indicating a lack of furin downregulation activity in mouse GBP2. Given the evolutionary pressure for viruses to diversify and evade host innate immune responses, it is plausible that a mouse factor capable of inhibiting or degrading furin protein is expressed in mouse cells. Further investigation is warranted to identify such a mouse restriction factor.

Mouse furin exhibits similar substrate specificity to human furin. In a natural environment, MLV replicates in mouse cells, with its Env glycoprotein being processed by mouse furin. To assess the impact of mouse furin on MLV Env cleavage and entry, attempts were made to establish mouse NIH3T3 cells stably expressing shRNA targeting mouse furin mRNA. Regrettably, such cells did not materialize. Nevertheless, observations of the dependence of the SARS-CoV-2 spike protein on furin in both human cells and ACE2-receptor-transgenic mice suggest that mouse and human furins share identical substrate specificities [60]. Therefore, comparable results may be anticipated in mouse cells.

## 4. Materials and Methods

### 4.1. Cells

Human 293T, human TE671, and mouse NIH3T3 cells have been provided by Dr. H. Amanuma and maintained in our laboratory since its establishment. These cell lines were cultured in Dulbecco’s modified Eagle’s medium (D-MEM) (Fujifilm-Wako) supplemented with 8% fetal bovine serum (Capricorn Scientific, Ebsdorfergrund, Germania) and 1% penicillin–streptomycin (Sigma-Aldrich, Saint Louis, MO, USA) at 37 °C in 5% CO_2_ environment.

To construct TE671 cells expressing the E-MLV receptor, CAT-1, 293T cells were transfected with expression plasmids to construct a CAT-1-expressing, VSV-pseudotyped MLV vector using the Fugene transfection reagent (Promega, Madison, WI, USA). The culture supernatant of the transfected cells was changed to fresh medium to remove the transfection reagent 24 h after the transfection, and the cells continued to be cultured for an additional 24 h. The culture supernatant was inoculated into TE671 cells, and the cells were selected with puromycin (2 ng/mL), because the MLV vector also encodes the puromycin-resistant gene. The puromycin-resistant cell pool was used in this study.

### 4.2. A-MLV-Pseudotyped HIV-1-Based Vector

HIV-1 Gag-Pol-Tat-Rev (1 μg), A-MLV Env (1 μg), and firefly-luciferase-encoding HIV-1 vector genome (1 μg) expression plasmids were mixed with the Fugene transfection reagent (5 μL) in 100 μL transfection solution in a 6 cm culture dish. The mixture was added to 293T cells. The culture supernatant of the transfected cells was changed to fresh medium to remove the transfection reagent 24 h after the transfection. The cells continued to be cultured for 24 h. The culture supernatants were added to TE671 cells and cultured for 2 days. Cell lysates were prepared from the inoculated cells, and their luciferase activities were measured using a firefly luciferase activity measure kit (Promega) by a luminometer (ATTO, Amherst, NY, USA).

### 4.3. MLV-Based Vector

MLV Gag-Pol (TaKaRa Bio, Kusatsu, Japan) (1 μg) and Renilla-luciferase-encoding MLV vector genome [44] (1 μg) expression plasmids were added into 100 μL transfection solution together with an Env expression plasmid (1 μg) of ecotropic Moloney (E-MLV), polytropic Friend (P-MLV), amphotropic (A-MLV), or XMRV (X-MLV). Subsequently, 5 μL Fugene transfection reagent was added to the solution, and the mixture was transferred to 293T cells. The culture supernatant of the transfected cells was added to target cells, and the inoculated cells were cultured for 2 days. The luciferase activities of cell lysates were measured using a Renilla luciferase activity measure kit (Promega).

### 4.4. Western Blotting

Cell lysates were prepared from target cells and subjected to SDS-PAGE (Bio-Rad, Hercules, CA, USA). Proteins were transferred onto a PVDF membrane (Milipore, Burlington, MA, USA). The membrane was treated with an antibody: rabbit anti-TM antibody (provided by Dr. A. Rein), mouse anti-GFP antibody, or mouse antiactin antibody (Santa Cruz). When the membrane was treated with a rabbit antibody, it was then treated with HRP-conjugated protein G (Bio-Rad). When the membrane was treated with a mouse antibody, it was subsequently treated with HRP-conjugated antimouse IgG antibody (Bio-Rad). The antibody-bound proteins were visualized and quantitated with the ECL reagents (Bio-Rad) using a chemiluminescence scanner (LI-COR Biosciences, Lincoln, NE, USA).

### 4.5. Site-Directed Mutagenesis

MLV Env mutants were constructed using an oligonucleotide-directed dual amber-based site-directed mutagenesis kit (TaKaRa Bio). Nucleotide sequences of phosphorylated primers used for the mutagenesis are shown in Figure 1.

**Table 1 ijms-25-09987-t001:** Nucleotide sequences of primers used in site-directed mutagenesis.

Primers	Nucleotide Sequences
E-MLV FM	CTG TTT GAG AGA AAA ACC AAA TAT AAA AGA GAG CCG
X-MLV FM	CAA TTT GAG AGA AGC AAC AGA CAT AAA AGA GAG CCG

## 5. Conclusions

The different strains of MLV have different dependence on the furin-mediated cleavage of the Env glycoprotein due to the amino acid sequence at the cleavage site. Thus, the furin-dependent MLV strains show susceptibility to the host antiviral factor GBP2, because GBP2 inhibits furin expression.

## Figures and Tables

**Figure 1 ijms-25-09987-f001:**
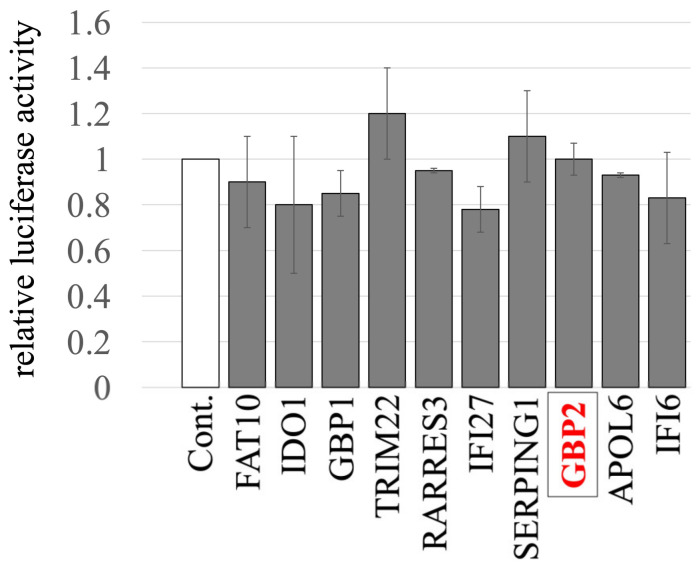
Impact of IFN-γ-inducible host factors on the late phase of the retroviral replication cycle. HeLa cells stably expressing one of the IFN-γ-inducible host factors were constructed using MLV vectors. Control cells were HeLa cells transduced with an empty MLV vector. These cells were transfected with plasmids to generate luciferase-encoding A-MLV-pseudotyped HIV-1-based vector. The supernatants from the transfected cells were inoculated into TE671 cells, followed by measuring the luciferase activity in their cell lysates. The luciferase activity of TE671 cells inoculated with supernatant from transfected control cells was always set to 1. This experiment was conducted in triplicate, and error bars show standard deviations. GBP2 is boxed.

**Figure 2 ijms-25-09987-f002:**
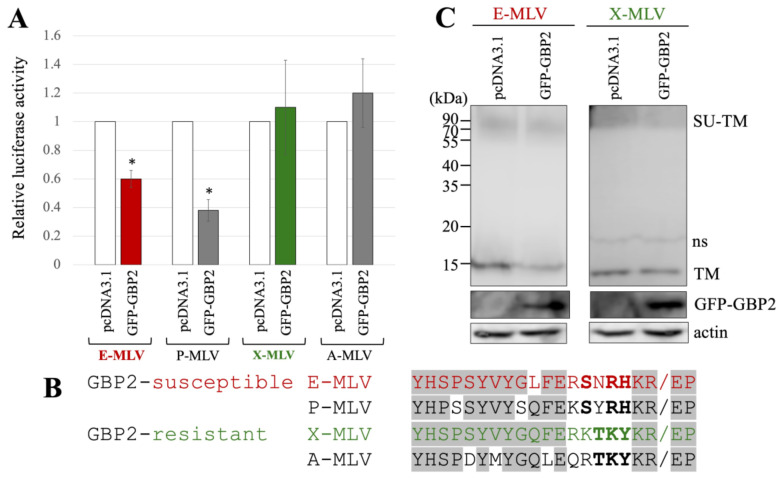
E-MLV and P-MLV are sensitive to GBP2, while X-MLV and A-MLV are resistant. (**A**) Human 293T cells were transfected with plasmids to construct the indicated MLV vector encoding luciferase together with either pcDNA3.1 or GFP-GBP2 expression plasmid. The culture supernatants from the transfected cells were inoculated into TE671/CAT1 cells, and luciferase activity was measured. The luciferase activity of cells inoculated with MLV vectors from pcDNA3.1-transfected cells was always set to 1. This experiment was conducted in triplicate, and error bars show standard deviations. Asterisks indicate significant differences compared to luciferase activity in pcDNA3.1-transfected cells. (**B**) Amino acid sequences near the SU-TM cleavage site of E-MLV, P-MLV, A-MLV, and X-MLV are indicated. Slashes show the cleavage sites. Bold letters indicate shared amino acid residues among GBP2-sensitive or -resistant MLVs. Letters in gray areas show shared residues among all MLVs. (**C**) Cell lysates from the transfected cells were analyzed by Western blotting using anti-TM, anti-GFP, or antiactin antibody. Non-specific bands are indicated as ns.

**Figure 3 ijms-25-09987-f003:**
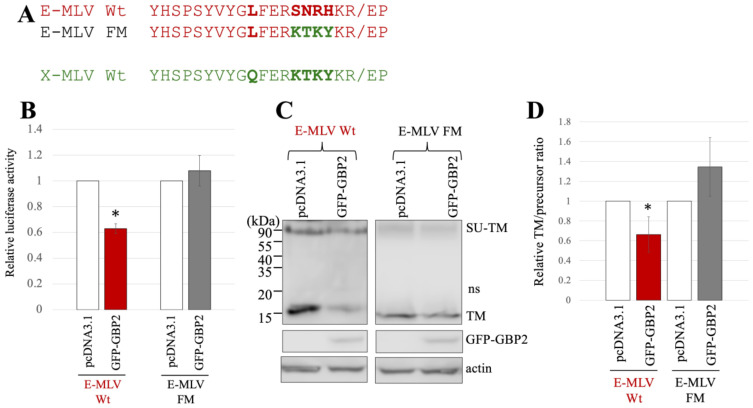
Substitution of the sequence near the cleavage site of E-MLV with that of X-MLV confers resistance to GBP2. (**A**) The sequence near the SU-TM cleavage site of E-MLV was substituted with that of X-MLV, resulting in the mutant designated as E-MLV FM. Amino acid sequences near the cleavage site of E-MLV Wt, E-MLV FM, and X-MLV Wt are indicated. (**B**) Human 293T cells were transfected with plasmids to generate E-MLV vectors containing the indicated Env proteins together with either pcDNA3.1 or GFP-GBP2 expression plasmid. Culture supernatants from the transfected cells were inoculated into TE671/CAT1 cells, and luciferase activities of their cell lysates were measured. The luciferase activity of cells inoculated with MLV vectors from pcDNA3.1-transfected cells was always set to 1. This experiment was conducted in triplicate, and error bars show standard deviations. Asterisks indicate significant differences compared to luciferase activity in pcDNA3.1-transfected cells. (**C**) Cell lysates from the transfected cells were analyzed by Western blotting using anti-TM, anti-GFP, or antiactin antibody. (**D**) Band intensities of TM and precursor Env were measured, and ratios of TM to precursor were calculated (*n* = 3). The ratio in pcDNA3.1-transfected cells was always set to 1, and error bars show standard deviations. Asterisks indicate significant differences compared to luciferase activity in pcDNA3.1-transfected cells.

**Figure 4 ijms-25-09987-f004:**
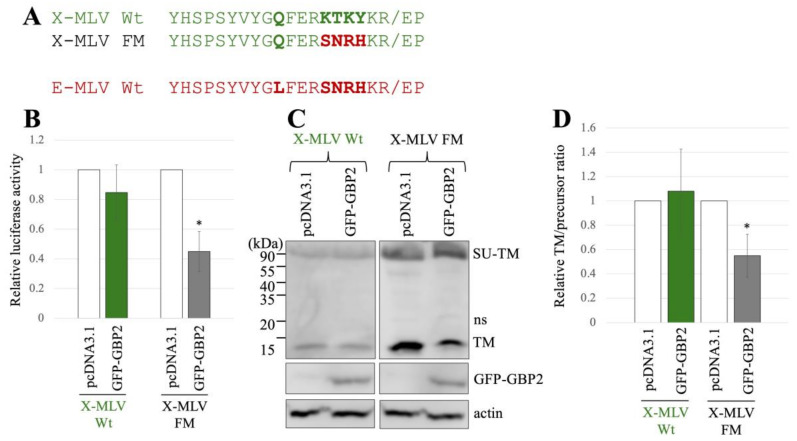
Substitution of the sequence near the cleavage site of X-MLV with that of E-MLV confers sensitivity to GBP2. (**A**) The sequence near the SU-TM cleavage site of X-MLV was substituted with that of E-MLV, resulting in the mutant designated as X-MLV FM. Amino acid sequences near the cleavage site of X-MLV Wt, X-MLV FM, and E-MLV Wt are indicated. (**B**) Human 293T cells were transfected with plasmids to generate X-MLV vectors containing the indicated Env proteins together with either pcDNA3.1 or GFP-GBP2 expression plasmid. Culture supernatants from the transfected cells were inoculated into TE671/CAT1 cells, and luciferase activities of their cell lysates were measured. The luciferase activity of cells inoculated with MLV vectors from pcDNA3.1-transfected cells was always set to 1. This experiment was conducted in triplicate, and error bars show standard deviations. Asterisks indicate significant differences compared to luciferase activity in pcDNA3.1-transfected cells. (**C**) Cell lysates from the transfected cells were analyzed by Western blotting using anti-TM, anti-GFP, or antiactin antibody. (**D**) Band intensities of TM and precursor Env were measured, and ratios of TM to precursor were calculated (*n* = 3). The ratio in pcDNA3.1-transfected cells was always set to 1, and error bars show standard deviations. Asterisks indicate significant differences compared to luciferase activity in pcDNA3.1-transfected cells.

**Figure 5 ijms-25-09987-f005:**
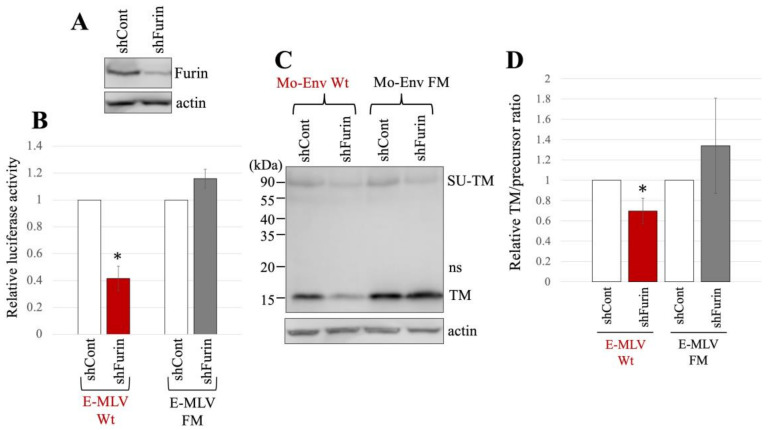
E-MLV Wt infection requires furin but E-MLV FM does not. (**A**) Human 293T cells were transduced with an HIV-1 vector encoding shRNA against furin (shFurin) or a scrambled sequence (shCont). Cell lysates from these cells were analyzed by western blotting using anti-furin or antiactin antibody. (**B**) Human 293T/shFurin or 293T/shCont cells were transfected with plasmids to generate E-MLV Wt or E-MLV FM vector. Culture supernatants from the transfected cells were inoculated into TE671/CAT1 cells, and luciferase activities of their cell lysates were measured. The luciferase activity of cells inoculated with MLV vectors from transfected 293T/shCont cells was always set to 1. This experiment was conducted in triplicate, and error bars show standard deviations. Asterisks indicate significant differences compared to luciferase activity in 293T/shCont cells. (**C**) Cell lysates from the transfected cells were analyzed by Western blotting using anti-TM, anti-GFP, or antiactin antibody. (**D**) Band intensities of TM and precursor Env were measured, and ratios of TM to precursor were calculated (*n* = 3). The ratio in 293T/shCont cells was always set to 1, and error bars show standard deviations. Asterisks indicate significant differences compared to the ratio in 293T/shCont cells.

**Figure 6 ijms-25-09987-f006:**
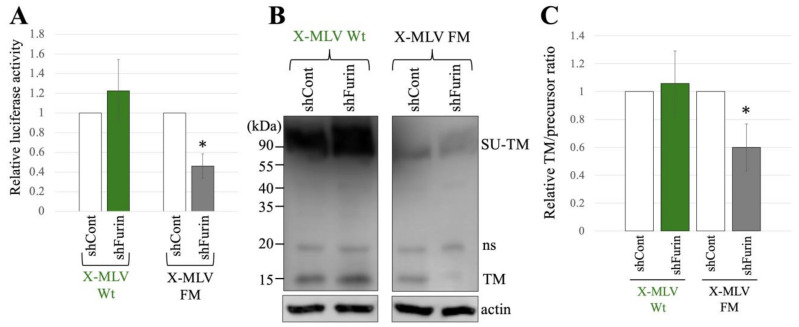
X-MLV FM infection requires furin but X-MLV Wt does not. (**A**) Human 293T/shFurin or 293T/shCont cells were transfected with plasmids to generate X-MLV Wt or X-MLV FM vector. Culture supernatants from the transfected cells were inoculated into TE671/CAT1 cells, and luciferase activities of their cell lysates were measured. The luciferase activity of cells inoculated with MLV vectors from transfected 293T/shCont cells was always set to 1. This experiment was conducted in triplicate, and error bars show standard deviations. Asterisks indicate significant differences compared to luciferase activity in 293T/shCont cells. (**B**) Cell lysates from the transfected cells were analyzed by Western blotting using anti-TM, anti-GFP, or antiactin antibody. (**C**) Band intensities of TM and precursor Env were measured, and ratios of TM to precursor were calculated (*n* = 3). The ratio in 293T/shCont cells was always set to 1, and error bars show standard deviations. Asterisks indicate significant differences compared to the ratio in 293T/shCont cells.

**Figure 7 ijms-25-09987-f007:**
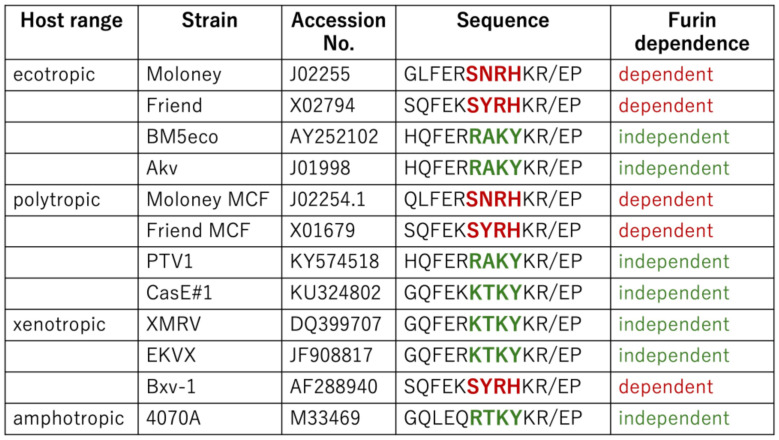
Host ranges and furin dependence of various MLVs. Amino acid sequences of various MLV strains are indicated. Furin dependence is determined by the amino acid sequence near the SU-TM cleavage site. When MLV infection depends on furin, it is indicated by red words. If not, it is indicated by green words.

## Data Availability

All data are shown in this paper.

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
