# Peer review of "The Furin Protease Dependence and Antiviral GBP2 Sensitivity of Murine Leukemia Virus Infection Are Determined by the Amino Acid Sequence at the Envelope Glycoprotein Cleavage Site"

_ijms, 2024, doi:10.3390/ijms25189987_

Round 1

Reviewer 1 Report

Comments and Suggestions for Authors

In this manuscript, Kubo et al. describe how variations in the amino acid sequence at the cleavage site of the Env glycoprotein lead to differing levels of dependence on furin-mediated cleavage among different MLV strains. This fact expected, although no scientific proofs were published until now.

Here I write some concerns about this study:

- The title is very unspecific, instead of “murine gamma-retrovirus”, I would directly talk about murine leukemia virus.

- Line 34: "a MLV-derived vector" instead of "an MLV-derived vector" 

- Figure 1. I would have liked to see the results from this screening experiment using other MLV-pseudotyped vectors, i.e. E-MLV or P-MLV. In this manuscript, authors are reporting how GBP2 has a different restriction potential between E-MLV & P-MLV versus X-MLV & A-MLV, as shown in Figure 2. Would any of host factors shown in Figure 1 restrict also infection from E-MLV or P-MLV? This information would improve the overall quality of the paper, because it would show first a bottom-up approach where multiple factors are considered as host restriction factors, then research focuses on one of them. With the current information shown in Figure 1, there would be no reason to further investigate GBP2 since no change in the infection is reported. A logical link between Figure 1 and subsequent Figures is clearly missing.

- Lines 239-247: Authors generate furin knockdown 293T cells (referred as 293T/shFurin) using a shRNA, but a phenotype validation of this cell line is not shown before results in 5B. I would include a separate western-blot where validation of this cell line is provided before 

- Why the authors are transfecting cells with a plasmid enconding for GFP-tagged GBP2 is not clear to me. Is GFP used to confirm transfection via fluorescence microscopy? Most analyses are done in cell lysates, therefore there is no need to use GFP to identify single transfected cells for this type of analysis.

- None of the western-blots show a molecular weight reference to corroborate the molecular size of the bands detected. 

- Line 187: “was measured” instead of “were measured”

- There is no reference to Figure 5C in the text, this could be added in line 277

- Lines 345 - 348: the statements in these lines need a reference

Comments on the Quality of English Language

Some minor mistakes, as indicated in the review report.

Author Response

Comment 1: The title is very unspecific, instead of “murine gamma-retrovirus”, I would directly talk about murine leukemia virus.

Response 1: Thank you for the valuable comments. As per the reviewer’s suggestion, the titles was changes to “Furin protease-dependence and antiviral GBP2-sensitivity of murine leukemia virus infection is determined by the amino acid sequence at the envelope glycoprotein cleavage site”.

Comment 2: Line 34, “a MLV-derived vector” instead of an MLV-derived vector.

Response 2: As suggested by the reviewer, the sentences were revised (line 34 and 36).

Comment 3: Figure 1. I would have liked to see the results from this screening experiment using other MLV-pseudotyped vector, i.e. E-MLV or P-MLV.

Response 3: As the reviewer mentioned, we have already analyzed the impacts of these host factors on the transduction efficiency of MLV vector containing Env proteins from different strains. As a result, we identified a host factor that exhibits an interesting phenotype dependent on the retrovirus family among the 11 IFN-g-induced factors. Currently, we are investigating the molecular mechanism by which this factor inhibits the production of retrovirus particles. Therefore, we are unable to present the data at this time. We appreciate your understanding.

Comment 4: Line 239-247: I would include a separate western-blot where validation of this cell line is provided before.

Response 4: As the reviewer advised, the western blot is shown separately in Figure 5, and the following sentence was added (line 258-260).“The levels of furin protein in 293T/shFurin cells were indeed lower than those in 293T/shCont cells (Figure 5A).”

Comment 5: Why the authors are transfecting cells with a plasmid encoding for GFP-tagged GBP2 is not clear to me.

Response 5: Due to budget constraints, we aimed to avoid the additional purchase of an anti-GBP2 antibody by constructing a GFP-GBP2 expression plasmid. Since we have already acquired an anti-GFP antibody, this approach allowed us to detect GBP2 protein expression. Importantly, it has been demonstrated that GFP-tagging does not affect antiviral activity of GBP2 [31]. We appreciate your understanding to our approach.

Comment 6: None of the western-blots show a molecular weight reference to corroborate the molecular size of the bands detected.

Response 6: As suggested by the reviewer, molecular wight reference was added to Figures 2, 3, 4, 5, and 6.

Comment 7: line 187: “was measured” instead of “were measured”.

Response 7: As suggested, it was changed to “was measured” (line 213).

Comment 8: There is no reference to Figure 5C in the text.

Response 8: As mentioned, “Figures 5C and D” was added in line 286.

Comment 9: Line 345-348: the statements in these lines need a reference.

Response 9: As suggested, a reference was added [54] (line 350).

Reviewer 2 Report

Comments and Suggestions for Authors

This study presents a detailed investigation into the mechanism by which the host restriction factor GBP2 suppresses the replication of the ecotropic Moloney murine leukemia virus (E-MLV). Specifically, the research elucidates how GBP2 influences the cleavage of the viral envelope glycoprotein (Env) at the SU-TM site.  This research provides significant insights into the molecular mechanisms governing MLV envelope cleavage and the role of GBP2 in viral restriction. The experimental design is sound, the data are compelling, and the conclusions are well-supported by the results.

For these reasons, I recommend the acceptance of this paper.

Minor comment:

 Line 62: “specie-specificity” should be replaced with “species-specificity”.

Author Response

Comment 1: Line 62: “specie-specificity” should be replaced with “species-specificity”.

Response 1: Thank you for the comment. As the reviewer mentioned, it was changed to “species-specificity” (line 62).

Round 2

Reviewer 1 Report

Comments and Suggestions for Authors

The authors have made signficant changes to the manuscript as well as answering all my questions.